# Horse Jumping and Dressage Training Activity Detection Using Accelerometer Data

**DOI:** 10.3390/ani11102904

**Published:** 2021-10-07

**Authors:** Anniek Eerdekens, Margot Deruyck, Jaron Fontaine, Bert Damiaans, Luc Martens, Eli De Poorter, Jan Govaere, David Plets, Wout Joseph

**Affiliations:** 1WAVES-IMEC, Department of Information Technology, Ghent University-IMEC, 9052 Ghent, Belgium; margot.deruyck@ugent.be (M.D.); luc1.martens@ugent.be (L.M.); david.plets@ugent.be (D.P.); wout.joseph@ugent.be (W.J.); 2IDLab-IMEC, Department of Information Technology, Ghent University-IMEC, 9052 Ghent, Belgium; jaron.fontaine@ugent.be (J.F.); eli.depoorter@ugent.be (E.D.P.); 3VETMED, Department of Reproduction, Obstetrics and Herd Health, Faculty of Veterinary Medicine, Ghent University, 9820 Merelbeke, Belgium; bert.damiaans@ugent.be (B.D.); jan.govaere@ugent.be (J.G.)

**Keywords:** dressage, show jumping, activity recognition, accelerometer, CNN, neural network, machine learning

## Abstract

**Simple Summary:**

Analyzing equestrian show jumping and dressage training movements can be greatly useful during training, but existing technologies fall short in terms of user convenience and detection of major horse training activities. As a result, attaching sensors to the horse’s legs could give a simple solution that is accessible to all riders. However, there is a scarcity of research on automatic classification of horse jumping and dressage training movements. Thus, the goal of this study was to use an advanced machine learning algorithm to categorize leg accelerometer data from the majority of dressage and jumping training motions. This is the first study to show that jumping and dressage training movements can be accurately identified and the velocity of different gaits and paces can be estimated with a minimal error.

**Abstract:**

Equine training activity detection will help to track and enhance the performance and fitness level of riders and their horses. Currently, the equestrian world is eager for a simple solution that goes beyond detecting basic gaits, yet current technologies fall short on the level of user friendliness and detection of main horse training activities. To this end, we collected leg accelerometer data of 14 well-trained horses during jumping and dressage trainings. For the first time, 6 jumping training and 25 advanced horse dressage activities are classified using specifically developed models based on a neural network. A jumping training could be classified with a high accuracy of 100 %, while a dressage training could be classified with an accuracy of 96.29%. Assigning the dressage movements to 11, 6 or 4 superclasses results in higher accuracies of 98.87%, 99.10% and 100%, respectively. Furthermore, during dressage training, the side of movement could be identified with an accuracy of 97.08%. In addition, a velocity estimation model was developed based on the measured velocities of seven horses performing the collected, working, and extended gaits during a dressage training. For the walk, trot, and canter paces, the velocities could be estimated accurately with a low root mean square error of 0.07 m/s, 0.14 m/s, and 0.42 m/s, respectively.

## 1. Introduction

Horse riding has become increasingly popular, and there is ample evidence that riders derive a lot of pleasure and enjoyment from engaging in various equestrian disciplines [1]. The most popular equestrian events are also Olympic-level disciplines, i.e., show jumping and dressage [2]. During training and competition a jumping horse completes a course with fences and a dressage horse performs different predefined gaits, patterns and movements. Detecting those movements during a riding session provides insights into training and is needed to quantify the type and intensity of training sessions. The fitness level of a horse is also an important determinant of injuries and therefore, monitoring of training sessions may have important added value in the assessment of performance ability and potential future injuries in a sport horse [3]. Analyzing locomotion data could not only provide insights in horses’ fitness level but also define horse qualities. For example, dressage horses that were higher placed in the Olympic Games had higher speeds in the extended canter due to taking longer strides [4].

Currently, the assessment of training and competition is performed mainly by trainers or riders analyzing their cellphone video recordings, as available technologies remain limited [5]. However, this approach poses many challenges. Firstly, it requires someone to film qualitatively; secondly, visual analysis requires time and experience; and thirdly, there is a lack of quantification and consistent objective assessment which makes it difficult to compare training sessions. To tackle these issues, a small accelerometer device attached to the horse tendon boots can provide a more general solution, available to all riders.

The problem with existing technologies is that they only address gaits and/or often rely on a daunting hardware setup [5]. As in this study, accelerometers are used previously to detect the gaits of the horse or to quantify the physical activity [5,6]. For example, Ref. [7] uses data of horses equipped with seven inertial measurement unit (IMU) sensors at high sampling rates to classify eight different types of gaits. Another study [5] uses a simple setup (e.g., smartwatch) but only detects walk, trot and canter movements of the horse and the canter lead is not detected. Detecting canter lead is important to balance the time spent on each canter hand and landing on the right canter after a jump helps both horse and rider to remain in a calm, balanced and controlled rhythm while making the horse quicker on turns and lead to a faster time. Moreover, it was found that horses approaching a fence with the incorrect canter lead were 5.9 times more likely to score faults [8]. Also, in the sport of dressage, detecting the hand on which a horse is conducting dressage exercises is critical for balancing the amount of time spent on each side during a dressage training session. However, the side of dressage movements has never been automatically identified before. Also, during a dressage course a significant percentage of the total score at all levels of competition is made up of different types of a gait (collected, working, medium and extended) [9]. Those various types are distinguished both stylistically and by speed of movement but there are no studies so far that have explored if those could be automatically detected. Moreover, advanced dressage movements such as passage and piaffe account for more than 25% of the overall score in both the Grand Prix and Grand Prix Special tests but have never been before automatically classified [10]. To the authors’ knowledge, this is the first study to show automatic detection, based on accelerometer data gathered by an off-the-shelf device, of the majority of jumping and dressage training activities (i.e., walk, trot, canter, jump, flying change, paces, piaffe, passage, shoulder-in, haunches-in, leg yield, half pass and pirouette) based on accelerometer data.

In this study, we use leg accelerometers to extend our previous algorithmic approach [11] to classify horse activity, to investigate if we could detect gaits, the canter lead, jumps, and flying changes during a jumping training. We also examined into whether we could classify gaits, paces, gait variations, and dressage movements during a dressage training. Finally, the elaboration of a velocity estimation model obtained from experimental velocity measurements enables us to complement important training metrics such as velocity, stride length and step frequency.

## 2. Related Work

In recent decades, a wide range of approaches have emerged for monitoring horse training. Electromyography, for example, is a technique used in electrodiagnostic medicine for evaluating and recording the electrical activity produced by muscles. It can be used to determine muscle recruitment and fatigue in horses [12]. Infrared thermography, the technique of acquiring and analyzing heat data from non-contact thermal imaging equipment, has been used to aid in diagnosing lameness [13]. There has already been research into distinguishing the movements of various horses using various ways. Existing research mostly focuses on gaits, lameness, or differentiating between activities based on thresholds. There is no in-depth study that the authors are aware of that especially focuses on a user-friendly technique for robustly monitoring dressage and jumping training. Table 1 gives a brief overview of related previous studies. There are three main categories: accelerometer and gyroscope solutions, video imaging solutions, and strain gauge solutions.

The research cited, however, do not necessarily dig deeper into a wide range of movements. Speed of various gaits is estimated in [16,17], coarse activities (standing, grazing and amublating) are identified in [6], while just various gaits (walk, trot and canter) are explored in [5,15]. Reference [7] recognizes eight different gaits, but the sensor placement and number are less user-friendly, and greater sampling rates (200–500 Hz) are employed, which is inefficient in terms of energy use. For classification, Refs. [6,15] use threshold-based approaches based on accelerometer data which allowed the determination of gaits by definition of distinct acceleration value ranges for stand, walk, trot and canter or differentiation between standing, grazing and ambulating in horses, respectively. In [4,9,10,19] temporal variables such as stride duration, suspension, and stance duration are extracted from video data and then used to determine which variables are sufficient to distinguish various paces (collected, working, and extended) or gait variations (piaffe and passage). As a result, no signal patterns are learned, making this approach less resilient against data from unknown breeds, as seen in [15], where several thresholds are required to detect gaits for different breeds of horses, ponies, and Icelandic horses. Therefore, deep learning algorithms are more accurate because they can recognize patterns and handle noisy or missing data. For example, Ref. [5] employs a neural network to classify walk, trot and canter based on features extracted from accelerometer data such as average peak-to-peak amplitude, variance and the time interval between consecutive crossings of the horizontal axis and [18] uses a neural network to classify lameness by using the Fourier-transformed head motion extracted from video data. While using features as input can aid in classification speed and reduce computing complexity, the drawback is that information contained in the raw data is lost. Additionally, some studies classify hoof strain gauge data for gait analysis using neural networks. For example, Ref. [21] explores the relationship between hoof wall deformation and ground reaction forces using a neural network. The technology for analyzing movement patterns and the forces associated with them has made substantial advances in the understanding of locomotion. Ref. [20] classifies strain gauge data in order to determine whether an animal is shoed, the speed at which it moves, and whether it is walking with a healthy gait. Refs. [22,23] study the usage of an artificial neural network to predict bone loading using strain and displacement measurements. To achieve our goal, we used a convolutional neural network, which has the advantage of automatically extracting features from raw accelerometer data using powerful computing capabilities.

## 3. Materials and Methods

### 3.1. Animals and Activities

Between July 2020 and January 2021, data were collected at various Belgian equestrian sport stables from 14 adult warmblooded sporthorses at different dressage or jumping levels. All the details about the subjects can be found in Table 2. This variety of horses is appropriate for our research as the variation in levels will contribute to the generalization of the machine learning model, as accelerometer data patterns will be different for different disciplines and levels. The horses’ dressage levels are preliminary, medium, advanced medium, intermediate, and grand prix, in order of rising difficulty. The horses’ jumping levels range from 1.10 m to 1.50 m, which represents the height of the fence during a jumping course. The exercising during data recording is carried out by the owners or familiar riders at their local training arena with a track surface of sand mixed with GEOPAT polyflakes [24].

Interesting activities performed for this research are gaits (walk, trot and canter), specific dressage and jumping training movements as described in the next sections.

#### 3.1.1. Dressage

The most important dressage activities during a training session or competition and classified in this study are divided into four categories: gaits, paces, variations on gaits, and movements, all of which are described according to the rules of the Fédération Equestre Internationale (FEI), the governing body for the equestrian sports as listed below [25,26].

**Halt:** a stop of all movement with all four feet equally balanced underneath the horse.
**Gaits**
**Walk:** a four-beat gait with footfalls following one another.**Trot:** a two-beat diagonal gait where diagonal pairs of legs move forward simultaneously with a moment of suspension between each beat.**Canter:** a three-way gait that begins with the inner hind leg, goes to the front with the outer fore leg, then the inner foreleg, also known as the lead leg, and ends with a moment of suspension.
**Paces**
**Collected:** a gait in which the horse’s neck is raised and arched, and the hocks are well engaged but with shorter steps than the other paces.**Working:** a gait between the collected and extended trots.**Extended:** a gait where the horse lengthens the steps to their maximum length due to strong impulsion from the hindquarters.
**Variations on the Gaits**
**Passage:** a very collected trot characterized by high knee lifts and hock movement.**Piaffe:** a passage done on the spot.**Flying change:** a change of canter lead in one stride with front and hind legs changing simultaneously at every fourth, third, second or every stride.
**Movements. Figure 1 shows the different movements.**
**Shoulder-in:** a three-track movement in collected trot where the horse bends around the inside leg of the rider, away from the movement direction. The inside front leg crosses over in front of the outside front leg, while the inside hind leg stays on the same track as the outside front leg.**Haunches-in:** a four-track movement in collected trot or canter where the horse bends around the inside leg of the rider, towards the movement direction. The front legs and shoulders remain on the original track.**Leg yield:** a lateral movement in working trot which a horse moves forward and sideways at the same time, with the inside legs crossing in front of the outside legs. The horse’s body is relatively straight or with a slight flexion of the poll away from the travel direction.**Half pass:** a leg-yield like movement but the horse is bent towards the direction of travel. The half pass is performed in collected trot or canter.**Pirouette:** a movement which is usually performed at the collected walk or canter where the horse’s forehand circles around the hind limbs while slightly bent in the direction of travel.

The riders all competed at Prix St-George level or higher. Horses performed one of the following or all main dressage movements according to their level after a thorough warm up. We also want to classify various gaits each with their own speed and style and in order of increasing speed, the following investigated are collected, working, and extended gaits. Therefore, recordings were made of the ponies and horses moving back and forth on a part of the long side of the arena, making collected, working and extended movements of each gait type in a random order. The camera was set up with its axis perpendicular to the horses’ line of motion. This line of motion had a length of 10 m and poles in the ground indicated the beginning, middle and endpoint of this line so that the velocity could be detected based on the video data.

#### 3.1.2. Jumping

The most important jumping activities during a training session or competition, classified in this study are walk, trot, left -and right canter, jump and flying change. Figure 2 illustrates the course that the horses used for this study completed after warming up for 15 min individually. The warm-up consisted of 10 min walking, trotting and cantering, followed by five 0.40 m high vertical jumps. All animals jumped several times a course of 5 total efforts in a random manner after the warm-up (2 verticals, 2 parallel oxers, and 1 double combination). At first the obstacle height was 0.90 m. Afterwards, this was increased for two out of three horses to 1.10 m. A flying change was performed if the horse landed in the wrong canter after the jump.

### 3.2. Data Collection

The device used in this work is the Axivity AX6 6-Axis Logging Device [27]. As illustrated in Figure 3, the device with dimensions 23 × 32.5 × 8.9 mm and a weight of 11 g is fitted to the lateral side of the tendon boots using VELCRO stick-on circles and tape. The devices were attached to the front tendon boots for all of the subjects. In addition, for horse 4, accelerometers were also connected to the hind tendon boots. An attachment convention for device orientation aids in the collection of consistent and comparable datasets. Positive x, y, and z-direction point towards the ground, forward along the horse, and inwards the horse’s leg, respectively, when the horse stands still.

### 3.3. Data Processing

This work uses a semi-automatic approach to alleviate the burden of manual annotation. Observations on the horses’ behaviors were made with video recordings when collecting data from the sensors. Part of the data is labeled using ELAN video recordings, where annotations can be made by choosing the duration of the segment where the action is performed and typing the annotation [28,29,30]. The model is trained on this data subset and new data prediction labels are generated, which are then manually corrected in ELAN. The ground truth is obtained by keeping the training samples while removing moments when the horse was out of the field of view of the camera or showing undesirable behaviors like spooking, protest, and so on. Because the raw signals from accelerometers are collected continuously, segmentation is performed on these raw signals, which are divided into windows of 0.5 s to 2.4 s depending on the optimal settings at each stage of the classification. In some stages of the classification, a Fixed-size Overlapping Sliding Window approach is used, which is known to produce superior results [31].

### 3.4. Datasets

A total of 13 datasets (3 jumping and 10 dressage trainings) with a sampling frequency of 50 Hz were collected from 14 different horses. There are a total of 372,339 samples. Of these samples, 58% are from a dressage training and 42% are from a jumping training. After 12 min of training, horse 1’s right sensor was replaced by a leg contact, and therefore the dataset could only be used partially. The right leg sensor file for horse 6 was corrupted, making this dataset useless. Table 3 lists the considered activities in this study with the proportion and absolute number of samples measured and the subjects performing the activity for the dressage and jumping measurements. The FEI limits dressage movements to only several gaits, but because they are useful for training purposes, we tested some of them in other gaits as well, indicated with an asterisk in the table.

### 3.5. Classification Process and Feature Extraction

As shown on the left of Figure 4, the jumping model consists of one phase, aiming at classifying walk, trot, left-canter, right canter, jump and flying change. For the creation of a high performance dressage model, a multi-phase approach was developed, aiming at classifying the gaits, the paces, the variations on the gaits and main dressage movements. The right side of Figure 4 displays the complete overview for the proposed approach. The first phase classified horse movement, pace and variations on the gait at a specific gait, distinguishing between halt, walk, trot and canter (phase 1 in Figure 4). Once a horse’s gait class is classified, a second classifier is applied in order to classify the direction a horse is moving in (phase 2 in Figure 4). Upfront knowledge of the hand on which the horse is exercising in, is only required for certain movements (i.e., shoulder-in and haunches-in walk and trot). Once a horse’s gait class is identified as walk, a classifier is applied in order to classify pirouette in walk and a superclass containing all the other walk movements and paces (phase 3 block 1 in Figure 4). The walk movements can then be further subdivided in lateral walk movements and paces (phase 3 block 2 in Figure 4). Finally, the lateral walk movements are subdivided in shoulder-in, haunches-in and half pass (phase 3 block 3 in Figure 4). Once a horse’s gait class identified as trot, a classifier is applied in order to classify various superclasses (phase 3 block 4 in Figure 4), i.e., the trot paces, variations on the trot gait and lateral trot movements. Next, the variations on the trot gait are subdivided in passage and piaffe (phase 3 block 5 in Figure 4). The lateral trot movements are further classified using three classes i.e., the superset side movement (leg-yield and half-pass), shoulder-in and haunches-in (phase 3 block 6 in Figure 4). Finally, the side movements are further classified using two classes, i.e., leg-yield and half-pass (phase 3 block 7 in Figure 4). Once a horses’ gait class identified as canter, a classifier is applied in order to classify various classes and one superclass, i.e flying change, pirouette and lateral canter movements and paces (phase 3 block 8 in Figure 4). The lateral canter movements and paces are then further subdivided in canter paces, shoulder-in canter, haunches-in canter and half pass canter (phase 3 block 9 in Figure 4). Once the horses’ activity is classified as a pace, a classifier is applied in order to distinguish between collected, normal and extended (phase 4 in Figure 4).

#### 3.5.1. Activity Classification

Each activity classification block in phases 1, 2, and 3 uses a hyperparameter-tuned version of our previously released algorithms for classifying new activities, as well as an optimized time period [11]. For the deep learning, the libraries TensorFlow [32] and Scikit-learn [33] are used. It is a hybrid convolutional neural network (CNN), i.e., a CNN for local feature extraction combined with features that retain knowledge about the global time series form. To this end, the max-pooling layer output of the CNN is flattened and fused with additional features. For each time window, a set of feature characteristics was extracted from the acceleration signals that already demonstrated to be important for classifying animal activities [14,34,35]. The input vector of the hybrid CNN contains raw accelerometer data samples, i.e., the acceleration of the left (L) and right (R) leg in three directions ($a_{xL},a_{yL},a_{zL},a_{xR},a_{yR},a_{zR}$). The output layer produces a probability distribution over the class labels. To find the best network structure for identifying activities, various hybrid CNN models with different shapes were used in this study. Each network model has a different number of convolutional layers, and the activation function for each convolutional layer is a rectified linear unit (ReLU). A dropout with a probability of 0.55 was applied to the convolutional layers. The first convolutional layer is always followed by a zero-padding operation.

#### 3.5.2. Velocity Estimation

The velocity estimation model of phase 4 of the dressage model proposed in this study is using the same convolutional network as for the activity classification. To estimate the velocity we add eight variables as input to our neural network: the acceleration of the two legs in three directions and the autorcorrelation function of the acceleration in the x-direction ($auto_{xL},auto_{xR}$). The height of test subjects (*h*) was added as the only feature. The model is optimized for a low mean squared error, which means that it is punished for making greater errors in the differences between expected and actual values. These variables were chosen because many papers have illustrated that they are closely correlated with stride parameters [36]. The desired network output is the velocity (*v*).

#### 3.5.3. Pace Classification

The classification of paces during phase 4 of the dressage model, is accomplished by comparing and selecting the best decision regions from features for decision boundary detection algorithms. As a result, a Quadratic Discriminant Analysis (QDA) model was used to predict the gait pace [37]. The input vector of the QDA consists of a subset of selected features calculated from raw accelerometer data. Various features are considered, i.e., height at withers, velocity, stride duration, stride length. Afterwards, the most important features are identified, i.e., the height at withers and velocity for discriminating different paces.

### 3.6. Performance of the Classification

Algorithmic performance was evaluated using overall accuracy and confusion matrices [38]. The overall model accuracy is the number of true positive instances of all behavioral classes divided by the total number of test instances. Here a true positive is the number of instances where the algorithm correctly classified the activity using video observations as a reference. Finally, confusion matrices are used to evaluate the designed model where the diagonal elements represent whether the predicted test data label is equal to the true label, while off-diagonal elements are mislabelled by the classifier.

## 4. Results for Jumping

The model evaluation is accomplished by training the model on the data of four horses (12, 14, 2, 3) and by predicting activities of two horses (5, 13). The jumping horses landed almost always on the correct leg during jumping training, so only two flying changes were performed by the jumping horses. Flying changes are considered as an important jumping training activity and therefore, we added data of three dressage horses (two in the training set and one in the validation set) performing nineteen flying changes to test the model’s performance.Because the data is measured at 50 Hz and the optimal sampling rate in our previous work [14] was set at 10 Hz, the dataset is sub-sampled at this rate. Figure 5 illustrates the jumping training classification results using 2 s samples of accelerometer data sampled at 10 Hz. The number of successfully recognized instances for each class is represented by the confusion matrix’s diagonal elements. Unrecognized instances or recognition errors are represented by the off-diagonal elements. The darker blue the cell is colored, the better the recognition accuracy.

The overall classification accuracy is 100%. The predictions are not affected by the reduction of the sampling frequency to 10 Hz. To the best of authors’ knowledge, automatic jumping training classification based on accelerometer data has not been studied previously and so no comparison with literature could be made.

## 5. Results for Dressage

Since not all horses performed every dressage activity due to their different levels, and in order to obtain a generically applicable model that provides good results for horses of various levels, the experimental dataset is randomly divided into two disjoint sets: 66% training and 34% test data. Table 4 presents the classification accuracies for dressage activities as well as the overall classification accuracy of dressage trainings at different modes of specialization. The overall classification accuracy is determined using the proportion of dressage activities present in the dataset which can be found in Table 3.

As can be concluded from this table, the first mode accurately (100%) classifies the gait with which a horse is performing any movement. With a 97.08% accuracy, the second mode classifies the gait with details on the side of movement. Movements and paces are distinguished from one another at the third way of classification, with a classification accuracy of 99.10%. The third mode classification uses information derived from the first mode, but not the second, because direction information is not required for the third classification mode. Paces, lateral movements, variations in gait, and other movements are recognized with a 98.87% accuracy in the fourth mode of classification. In the last and most advanced mode of classification, all the recorded dressage movements are detected separately with an overall accuracy of 96.29%. The mean and median classification accuracy for all movements are 92% and 98%, respectively. Shoulder-in in walk and haunches-in in trot performing the worst, with classification accuracy of 69.35% and 71.60%, respectively. Halt, half pass in walk, and pirouette in walk are all classified with an excellent accuracy of 100%. The walk paces are distinguished from one another in this last way of classification. Group assignment to a velocity profile is necessary for the classification of walk paces, as will be discussed in Section 5.6.3. The different phases of classification as illustrated in Figure 4 from which these results arise are discussed in detail in the next sections.

### 5.1. Phase 1: Activity Group

Figure 6 shows the confusion matrix of the superclassifier for phase 1 classification when all the 25 dressage activities are merged into four disjoint sets to create supersets (i.e., halt, walk, trot and canter).

The overall classification accuracy is 100%. As expected, the activity superclassifier model presents good classification performances for halt, walk, trot and canter activities since one study found already that one could determine the gaits, without other activities, by means of individual acceleration value ranges [15].

### 5.2. Phase 2: Effect of Side

For the second phase of the classification, a 4 s, 1 s, and 2 s time interval were found to be the most optimal for classifying the side of the walk, trot, and canter movements, respectively. Figure 7 shows the confusion matrices for prediction of the direction of the walk, trot and canter activities of the superclassifier.

As expected, the direction superclassifier model performs well in classification for canter activities, as one study found that an Long Short-Term Memory (LSTM)—based model could identify the canter lead with an accuracy of more than 96.5% [7]. In our research, we are able to accurately (100%) classify not only the canter lead, but also the direction of other canter dressage activities (paces, shoulder-in, haunches-in, half pass, and pirouette). In addition, with an overall accuracy of 97.67%, the direction of walk activities can be classified almost perfectly. The model validated on the data of the trot activities performs the worst with a classification accuracy of 93.36% due to misclassification of the direction of shoulder-in, haunches-in, leg yield and half pass.

### 5.3. Phase 3: Classification of Walk Movements

Figure 8 shows the confusion matrix of the classifier at phase 3 block 1 from Figure 4. The overall classification accuracy is 100%.

To the best of authors’ knowledge, the automatic activity classification of lateral walk movements and paces and pirouette in walk has not been studied previously and so no comparison with literature could be made. Further, the superclass lateral walk movements and paces can then be subdivided. Figure 9 shows the confusion matrix for the classification of lateral walk movements and walk paces (phase 3 block 2). The overall classification accuracy is 99.35%.

The lateral walk movements are classified with an accuracy of 100% while only two samples of walk paces are misclassified as lateral walk movements leading to an accuracy of 99.32%. The latter class can then be further classified by the method as described in Section 5.6.

The lateral walk movements, i.e., haunches-in, shoulder-in and half pass are classified using the number of samples with a time interval of 0.64 s. Figure 10 shows the confusion matrix for the classification of the lateral movements in walk when the direction is unknown. The overall classification accuracy is 64.86%.

Something that stands out is the confusion between dressage movements shoulder-in, haunches-in and half-pass. As illustrated in Figure 1, these movements are defined by bending around the rider’s leg while traveling in a particular direction. Due to the fact that accelerometer data does not include information about the position of the legs, this could explain why the model is confusing those various movements. Therefore, we want to provide extra information to the model so that the accuracy for those movements can be improved. Since shoulder-in and haunches-in are always performed on the same side were the horse was previous in, we can add this information to the model. Since the direction of a half pass in walk is independent of the previous hand, no side information for this movement was added. The hand on which a horse is exercising in, can be determined with an accuracy of 97.67% as discussed in Section 5.2.

Figure 11 shows the confusion matrix for the classification of the lateral movements in walk when the direction is known for shoulder-in walk and haunches-in walk. The overall classification accuracy is 86.84%.

Adding information on the side on which the horse is walking leads to an overall increase of 22% in classification accuracy. Half pass walk is classified with an accuracy of 100%. Three occurrences of shoulder-in walk have been erroneously labeled as half-pass walk. This could be because both actions involve the crossing of the front legs, as explained by the definitions in Section 3.1.1. To the best of authors’ knowledge, classification of shoulder-in, haunches-in and half pass has not been studied previously and so no comparison with literature could be made.

### 5.4. Phase 3: Classification of Trot Movements

Figure 12 shows the confusion matrix of the trot superclassifier from phase 3 block 4 which distinguishes trot paces, variations on the trot gait and lateral trot movements. The overall classification accuracy is 98.08%.

One study showed that the stride duration determined from video data was enough to differentiate collected trot from passage and piaffe [10]. In our study, we are able to reliably (97.92%) differentiate variations on the trot gait from all trot paces (collected, working, and extended), as well as from other trot movements (shoulder-in, haunches-in, leg yield and half pass). The *trot paces* (collected, normal and extended) can then be further classified by the method as described in Section 5.6.

Figure 13 shows the confusion matrix for the classification of piaffe and passage (phase 3 block 5). The overall classification accuracy is 98.77%.

Passage is classified with an accuracy of 100% and only one sample of piaffe is misclassified as passage. This is as expected since one study found that the mean values of most temporal variables differed significantly between passage and piaffe [10]. The accelerometer patterns of the signal in the y-direction (forward direction) are more fluctuating for the passage movement than for the piaffe movement, as illustrated in Figure 14, since the horse stays more or less in one position during piaffe.

In phase 3 block 6, the lateral trot movements are distinguished from each other (shoulder-in, haunches-in and side movements). As in walk, shoulder-in and haunches-in in trot are always performed on the same side were the horse was previous in, we can add this information to the model. Since side movements such as half pass and leg yield may be ridden to either the left or right side, regardless of the horses’ previous direction, no side information is included in the model for those movements. As discussed in Section 5.2 the side of all the trot movements can be predicted with an accuracy of 93.36% due to misclassification of the side of shoulder-in, haunches-in, leg yield and half pass. During a dressage training session, shoulder-in and haunches-in excersises are always followed after a trot pace (mostly collected trot) [25]. Therefore, the side of only the trot pace was classified. A sliding time window approach is used in this step of the classification because it resulted in higher classification accuracies. As shown in the corresponding confusion matrix (Figure 15), the right side trot pace can be predicted with an accuracy of 100% while the left side, due to the misclassification of two instances, can be predicted with an accuracy of 96.22%. The overall side of the trot pace can be predicted with an accuracy of 98.31%.

Figure 16 shows the confusion matrix for the classification of the side movements (half-pass and leg-yield), haunches-in and shoulder-in when the side of the trot pace prior to the haunches-in and shoulder-in is known.

As the confusion matrices show, shoulder-in is classified with the highest accuracy on both the left and right sides. The class with the lowest classification accuracy is haunches-in, with a left and right side classification accuracy of 76% and 75%, respectively. In both cases, haunches-in is counfused with a side movement in trot. The side movements get good classification results (≥85%) in both cases. As previously stated, lateral movements all require the horse bending around the rider’s leg while travelling in a certain direction, which may explain the model’s confusion. To the best of authors’ knowledge, classification of the various lateral movements has not been studied previously and so no comparison with literature could be made.

#### Effect of Multiple Sensors

When considering training classification, the question of multiple sensors impacting on the classficiation accuracy is important, and more specifically what trade-offs if any might be needed in terms of acceptable accuracy versus number of sensors. Therefore, horse 4 wore hind leg accelerometers during exercise to examine the effect of more and other combinations of accelerometers on the classification accuracy of lateral side movements. Table 5 shows the accuracy of the classification algorithm for the considered behaviors when data from the front- and back-mounted accelerometers is combined and used for classification while no upfront side information was used for shoulder-in and haunches-in.

As seen in this table, accelerometer data from the left and right front legs results in the lowest classification accuracy for lateral trot movements. When using either two or four sensors, adding hind leg data improves classification accuracy to on average 79%. When diagonal leg data from the right front leg and left hind leg is combined, the maximum classification accuracy of 84.85% for lateral side movements is achieved.

Figure 17 shows the confusion matrix for the classification of the side movements in trot, i.e., leg-yield and half-pass. The overall classification accuracy is 95.74%.

As can be concluded from the confusion matrices, there is only one misclassification of left leg yield and one misclassification of left half pass. Furthermore, because there is no mixing of left and right movements, the direction of the movements can be accurately determined. The distinct differences in the movements can account for those excellent classification findings. Leg-yield is done while bending away from the direction of movement in the working trot, and half-pass is done while bending towards the direction of travel in the collected trot. To the best of authors’ knowledge, classification of the leg yield versus half pass has not been studied previously and so no comparison with literature could be made.

### 5.5. Phase 3: Classification of Canter Movements

Figure 18 shows the confusion matrix of the canter classifier at phase 3 block 8 in Figure 4. At this stage flying change, lateral canter movements and paces and pirouette canter are distinguished from each other. The overall classification accuracy is 99.36%.

As can be concluded from the confusion matrix, only one sample of flying change is misclassified as pirouette in canter.

The next phase of the classification (phase 3 block 9 in Figure 4) of the canter movements consists of classifying haunches-in, shoulder-in, half-pass and canter paces (collected, normal and extended). Figure 19 shows the confusion matrix and normalized confusion matrix for the classification. The overall classification accuracy is 98.43%.

As predicted, there is no mixing of left and right side movements because the direction could be distinguished with 100% accuracy. Haunches-in and left-shoulder-in are performing the worst with accuracies between 75% and 83%. Haunches-in gets misclassified as half pass and left shoulder-in canter gets misclassified as left-canter. The other ’canter classes’ get classified with high accuracies between 99% and 100%. More training samples will be required to improve accuracies even further. To the best of authors’ knowledge, shoulder-in, haunches-in and half pass canter are never before classified so no comparison with literature could be made. The *canter paces* (collected, normal and extended) can then be further classified by the method as described in Section 5.6.

### 5.6. Phase 4: Collected, Extended and Normal Gaits

Each pace is characterized not only by *style*, but also by the horse’s *velocity*. As a result, a velocity estimation model based on accelerometer data is needed. Therefore data of seven horses and ponies (height at withers ≤ 148 cm) were collected, measuring the velocity at each pace. The estimated velocity can be used as an input to calculate the stride length which is an important training metric according to horse riders. Stride length is related to velocity and stride duration and can be calculated as follows [4]:(1)L=v·ΔT
with *L* the stride length, *v* the velocity and ΔT the stride duration. The stride duration is the inverse of the step frequency (f=1/ΔT) and can be determined by the highest peak in the autocorrelation function of the acceleration in the x-direction. The peaks in the acceleration signal in the vertical direction reflect the impact moments. The autocorrelation function displays the correlation of a signal with itself as a function of delay [39]. As a result, the moment when signal rehearses itself is represented by the highest peak in the auto correlation function. Table 6 lists the mean measured velocities, the stride duration and length for the horses and ponies.

The mean velocities, stride duration and stride lengths of the horses are similar for the various types of gaits found in literature for horses [4,9,19]. In our study also two ponies are analysed. To our knowledge, there are no studies about the stride parameters of collected, working and extended gaits of ponies or the relationship between breed or height at withers and stride parameters. In all gaits, the horses and ponies’ speed increased from collected to working to extended pace. Increases in speed are usually accompanied by an increase in stride length. As walk speed increased, the stride duration appeared to decrease. The stride duration remained more or less constant as the speed increased in the trot and canter. The alternation of stride length and duration is irregular and there is no ready explanation for this. The ponies’ speed, stride length, and stride duration were usually lower than the horses’ in all gaits.

#### 5.6.1. Velocity Estimation

In Figure 20 the velocity estimation results are presented together with the root mean squared error (RMSE) and the $R^{2}$-value. The square root of the mean of the square of all errors is the RMSE, which is a commonly used measure of the variations between values expected by a model and the values actually observed. The $R^{2}$-value is a statistic that indicates the goodness of fit between the expected and observed values.The *x*-axis denotes the estimated velocity, the y-axis denotes the true velocity. The true and estimated velocity should theoretically match on a line with slope one, meaning that the estimated value equals the true value.

As seen in the figures, all velocities can be estimated fairly accurately, with the best estimations for walk paces and the worst estimations for canter paces (RMSE = 0.07 m/s and $R^{2}$-value = 0.92 vs RMSE = 0.42 m/s and $R^{2}$-value = 0.92). This is to be expected, considering that canter paces have more variability in velocity and there are fewer canter samples to train the model on because the velocity was higher but the distance covered remained the same. Another study found similar results when predicting speeds in five gaits (walk, trot, tölt, pace, and canter), with an RMSE increasing from 0.20 to 0.34 m/s as the algorithm assessed faster gaits [17]. We hypothesize that with additional canter training data, the estimated canter velocity will be more accurate.

#### 5.6.2. Decision Regions

The aim of phase 4 classification is to distinguish the various gait paces. As a result, now that the velocity can be estimated accurately, various decision regions for the paces need to be established. Various features, such as height at withers, velocity, stride duration, and stride length, are calculated, but feature selection revealed that height at withers and velocity are the most important features for distinguishing between different paces with the QDA classifier. Figure 21 shows the classification boundaries with QDA based on the heights of the horses together with the measured velocities for the paces in the gaits walk, trot and canter. The overall classification accuracy for walk, trot and canter paces are 69.70%, 100% and 100%, respectively.

As can be concluded from the decision boundary plots, the trot and canter paces can be predicted with an accuracy of 100% since for trot and canter there is almost no overlap between the speeds of the various paces of all the horses and ponies which makes it appropriate for decision boundary based classification. But fixed decision boundary-based techniques are not suitable for detecting the different types of walk since they can only be predicted with an accuracy of 69.70%. The fact that different horses’ paces overlap in walk, even in a group of horses of similar height at withers as shown in Figure 21a, highlights the importance of looking at individual velocity profiles rather than drawing conclusions based on velocity regions, which will be addressed in the following section.

#### 5.6.3. Group Velocity Profile

Although the walk velocity can be estimated with a low root mean square error of 0.07 m/s, different horses’ velocities and paces overlap. As a result, an individual or group velocity profile must be established, and an input is needed to address a horse to a specific group velocity profile. Therefore, in this study, three groups are created, i.e., ponies (height at withers ≤ 148 cm), slow horses and fast horses (minimum speed at collected walk ≥ 1.6 m/s).

Figure 22 shows the confusion matrix of the QDA classifier for the walk paces when three groups are considered. The overall classification accuracy for the ponies, slow horses and fast horses are (a) 100%, (b) 100% and (c) 87.5%, respectively.

As can be concluded from the confusion matrices, the walk paces of the ponies and slow horses can be predicted with an accuracy of 100% while the walk paces of the fast horses can be predicted with an accuracy of 87.5% due to the misclassification of one sample of extended walk as walk.

## 6. Conclusions

In this study we propose a solution for a horse jumping and dressage training activity recognition problem that is based on based on leg accelerometer data. The results show that for the first time 6 jumping training activities and 25 advanced dressage training movements using an experimental dataset from 14 different types of horses could be automatically classified with high accuracies, i.e., 100% and 96.29%, respectively. The experiment also demonstrated that various subjects’ velocities can be correctly estimated, resulting in a stride length estimation model. These results can help in further development of an automatic system for training activity detection and help improve training sessions and horses’ fitness levels. Capturing and analysing the medium variant of a gait will be included in future work. Our suggested approach demonstrates superior potential in most cases as shown by the above experimental results, but the main limitations of this study are the reduced number of horses for each training activity group with data of only two ponies present in our dataset. We conjecture that, with more training data of different breeds, our behavior detector will be more robust to these different cases.

## Figures and Tables

**Figure 1 animals-11-02904-f001:**
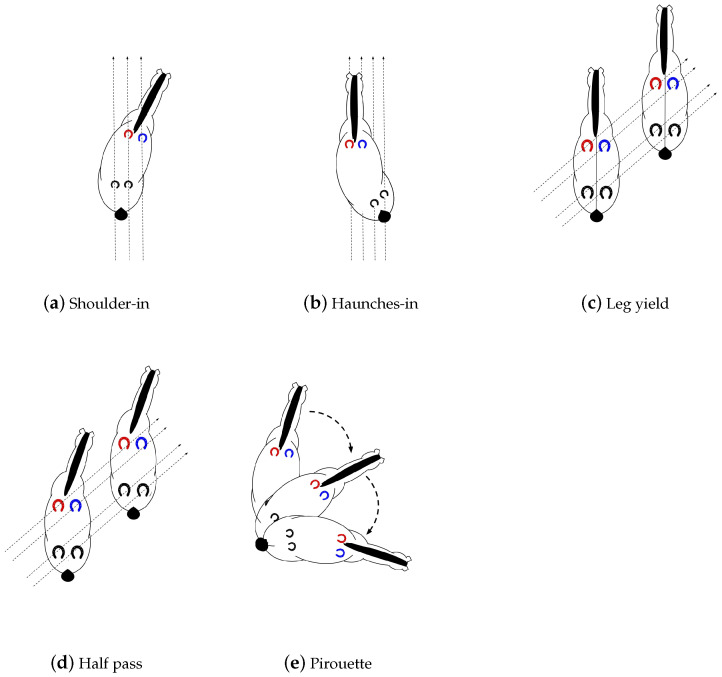
Classified dressage movements.

**Figure 2 animals-11-02904-f002:**
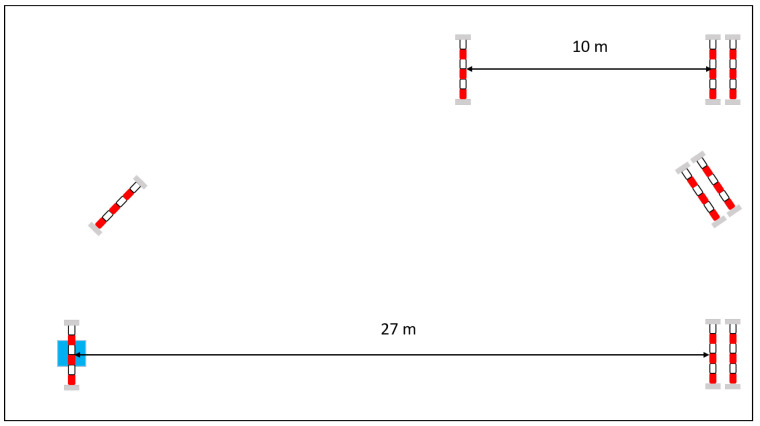
Jumping course with 5 efforts (2 verticals, 2 parallel oxers, and 1 double combination) with an obstacle height of at first 0.90 m and afterwards 1.10 m.

**Figure 3 animals-11-02904-f003:**
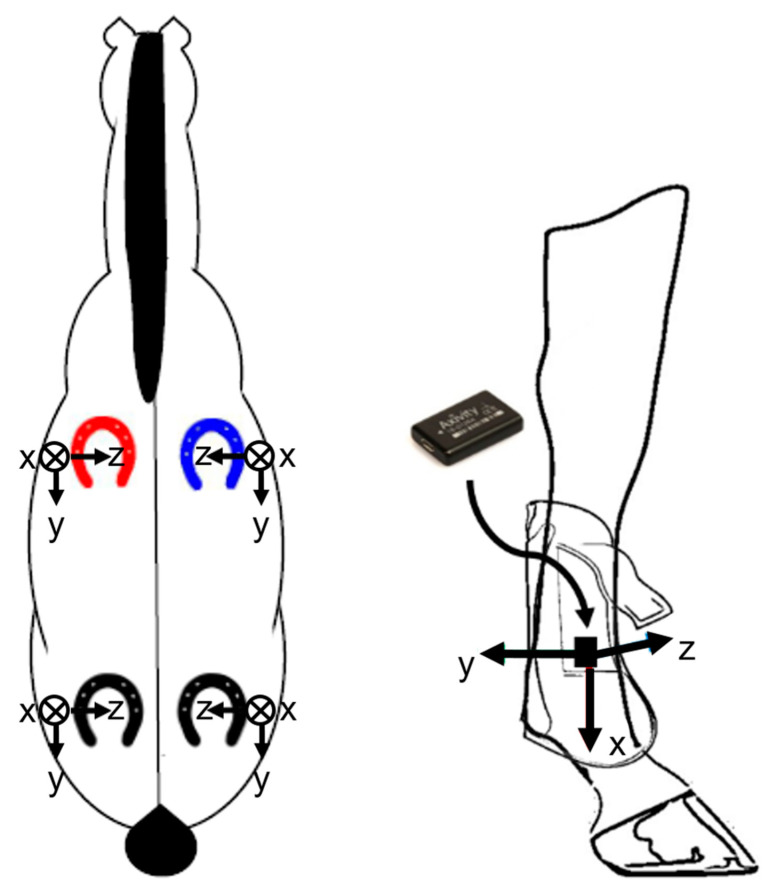
Position and orientation (*X*, *Y*, and *Z* axes) of the accelerometers.

**Figure 4 animals-11-02904-f004:**
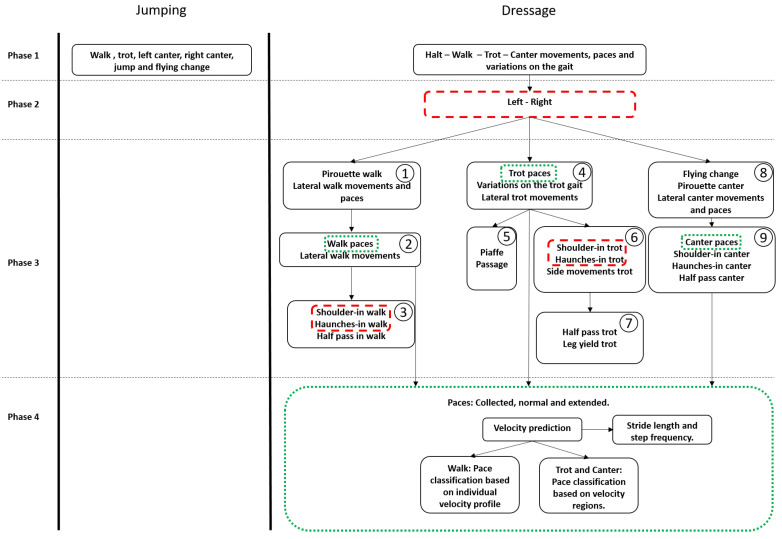
Overview of the jumping model (**left**) and dressage model (**right**).

**Figure 5 animals-11-02904-f005:**
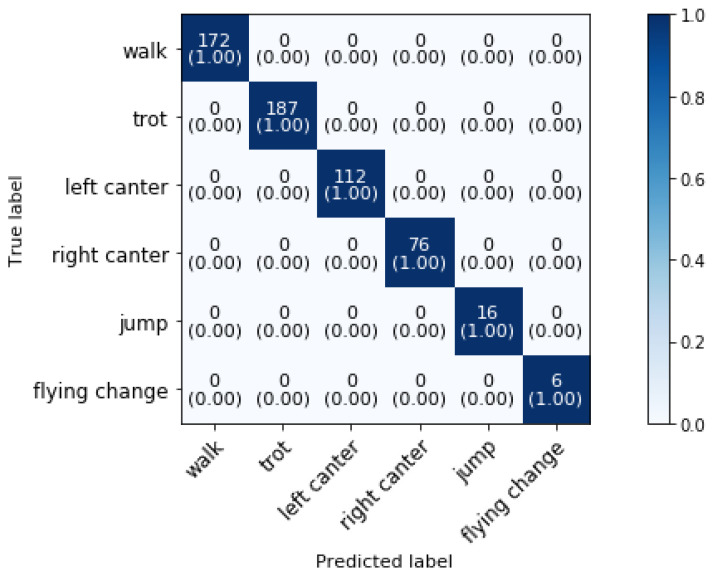
Confusion matrix for the validation set of a jumping training with a sampling rate of 10 Hz and a time interval of 2 s, achieving 100% overall accuracy.

**Figure 6 animals-11-02904-f006:**
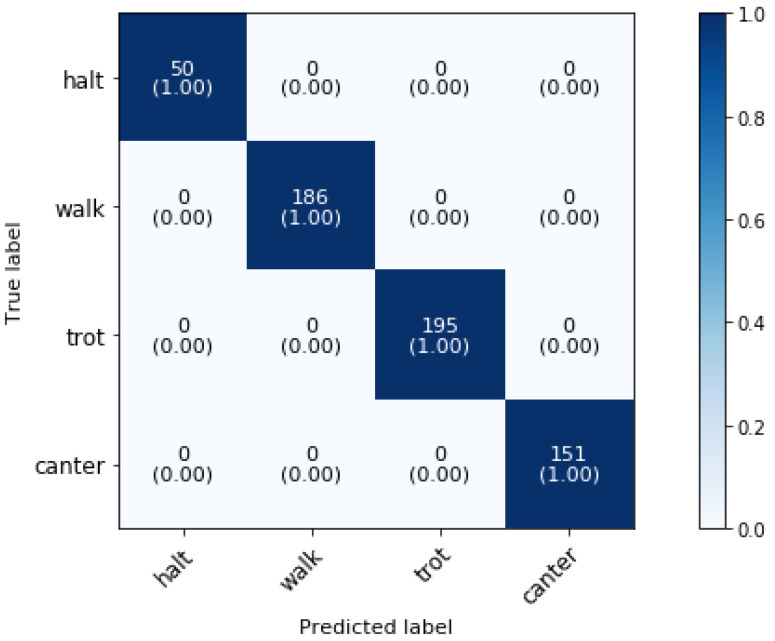
Confusion matrix for the validation set of a dressage training activity classification at phase 1 at a sampling rate of 50 Hz and a time interval of 2 s, achieving 100% overall accuracy.

**Figure 7 animals-11-02904-f007:**
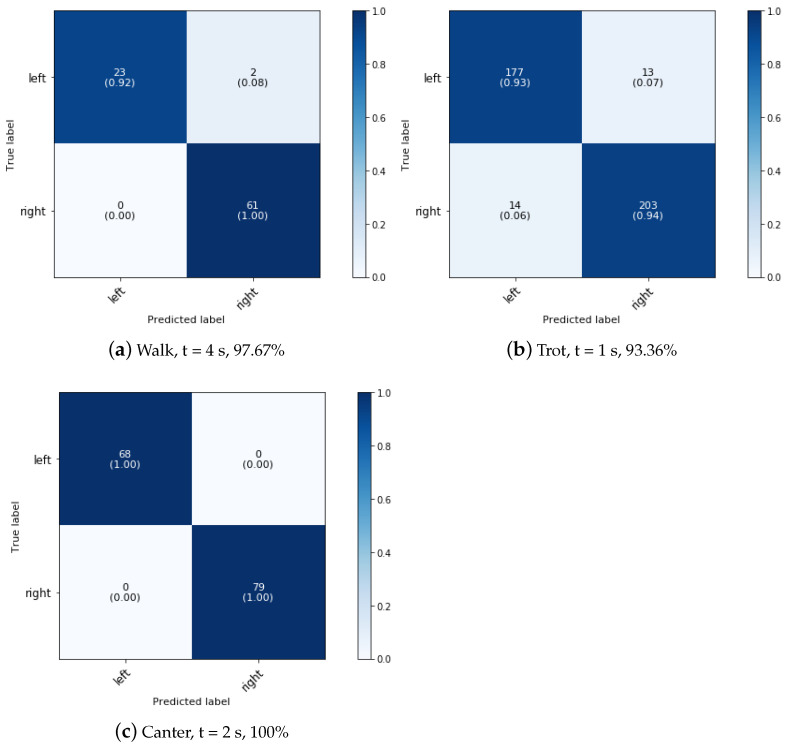
Confusion matrix for the validation set of a dressage training side classification at phase 2 at a sampling rate of 50 Hz for walk, trot and canter movements, paces and variations on the gait.

**Figure 8 animals-11-02904-f008:**
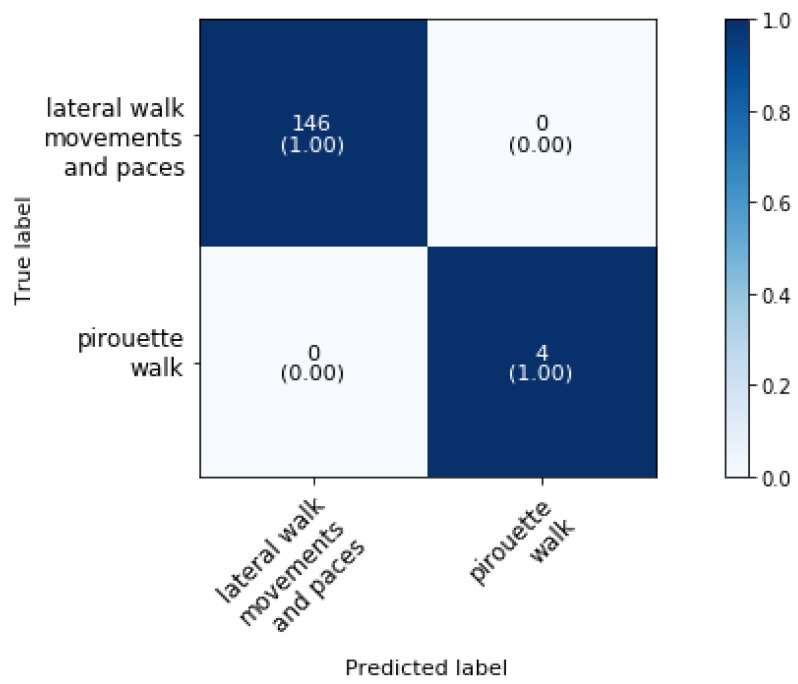
Confusion matrix for the validation set a dressage training classification at phase 3 block 1 at a sampling rate of 50 Hz and a time interval of 2.4 s, achieving an overall classification accuracy of 100%.

**Figure 9 animals-11-02904-f009:**
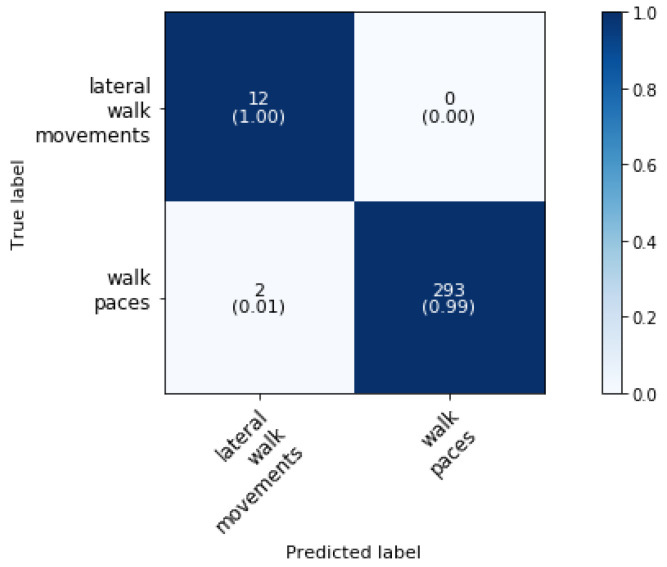
Confusion matrix for the validation set a dressage training classification at phase 3 block 2 at a sampling rate of 50 Hz and a time interval of 1.2 s, achieving an overall classification accuracy of 99.35%.

**Figure 10 animals-11-02904-f010:**
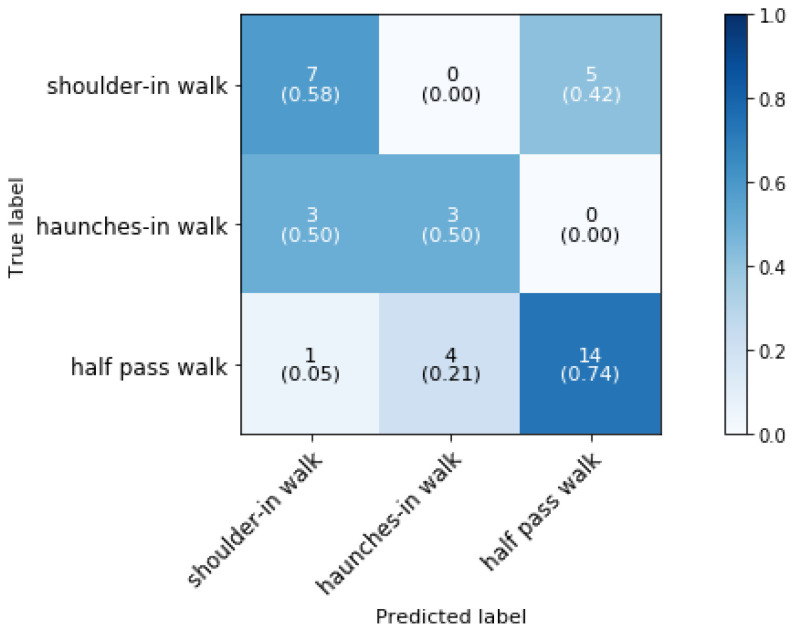
Confusion matrix for the validation set of phase 3 block 3 for an unknown direction in a dressage training at a sampling rate of 50 Hz and a time interval of 0.64 s, achieving 64.86% classification accuracy.

**Figure 11 animals-11-02904-f011:**
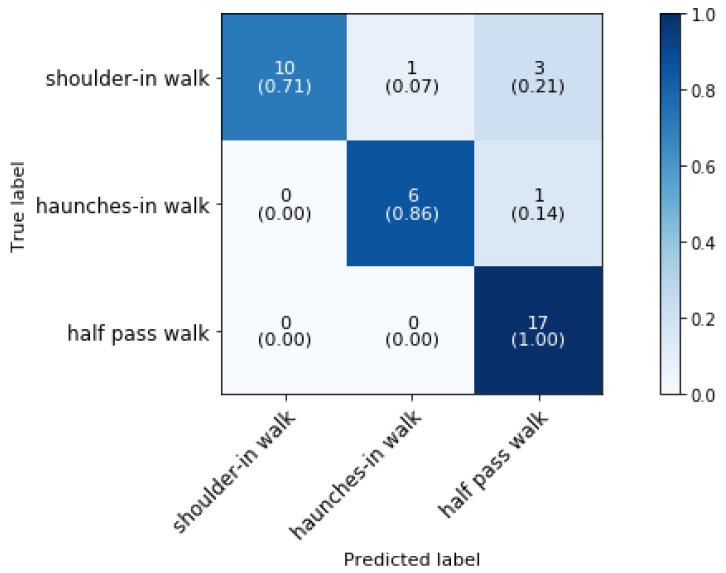
Confusion matrix for the validation set of phase 3 block 3 for a known direction in a dressage training at a sampling rate of 50 Hz and a time interval of 0.64 s, achieving 86.84% classificiation accuracy.

**Figure 12 animals-11-02904-f012:**
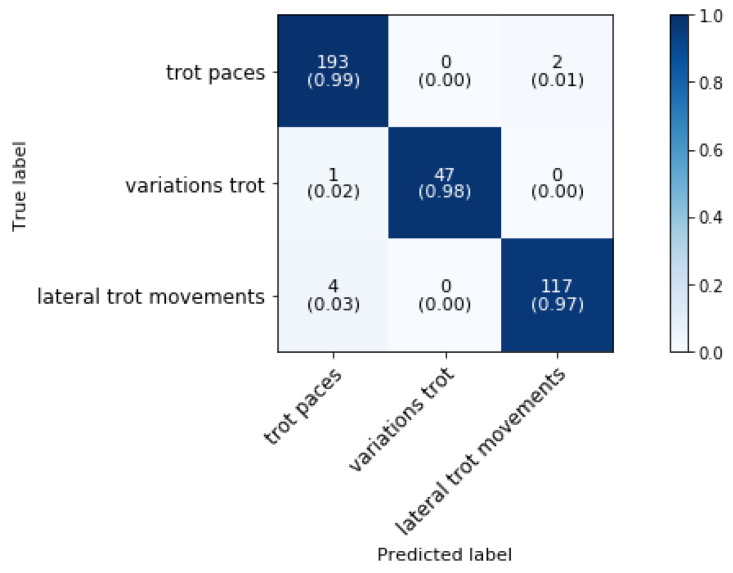
Confusion matrix for the validation set of the trot activity groups in a dressage training at a sampling rate of 50 Hz and a time interval of 0.8 s, achieving 98.08% overall accuracy.

**Figure 13 animals-11-02904-f013:**
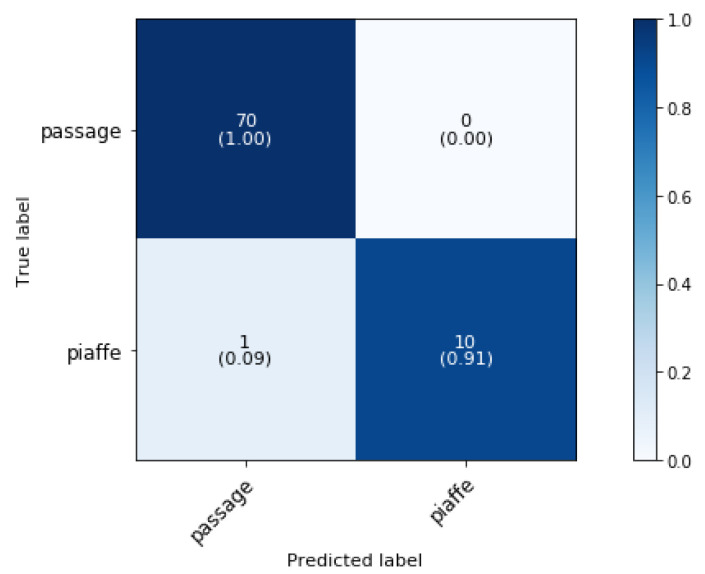
Confusion matrix for the validation set of the variations on the trot gait in a dressage training at a sampling rate of 50 Hz and a time interval of 0.5 s, achieving 98.77% overall classification accuracy.

**Figure 14 animals-11-02904-f014:**
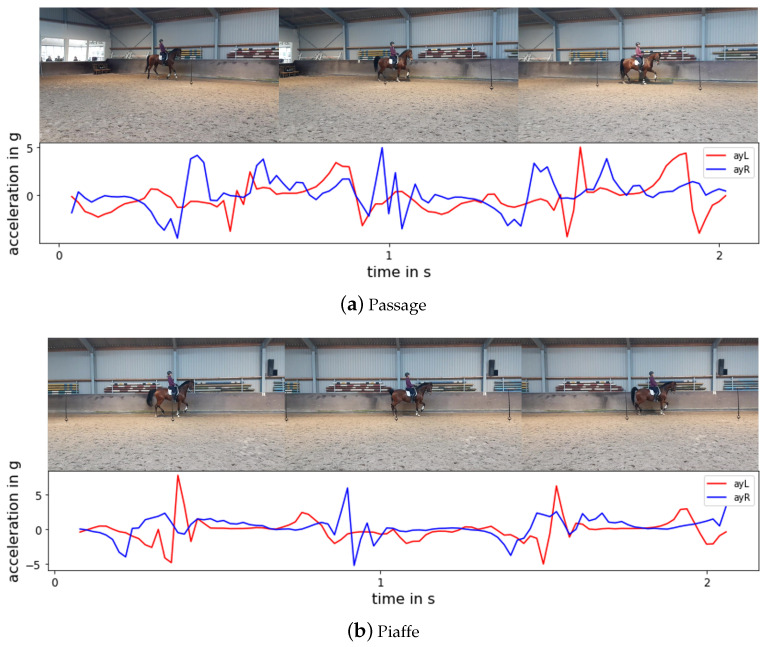
Typical accelerometer patterns of (**a**) passage and (**b**) piaffe in a 2 s window. The red line represents the Y signal from the left accelerometer and the blue line represents the Y signal from the right accelerometer, respectively.

**Figure 15 animals-11-02904-f015:**
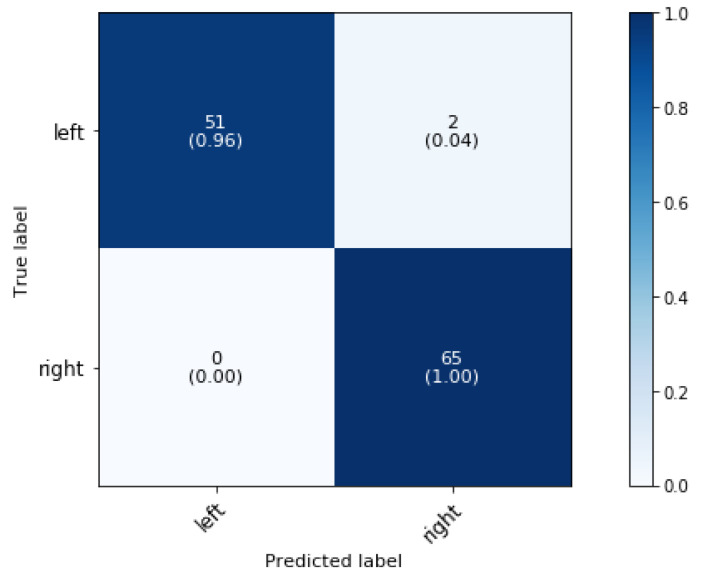
Confusion matrix for the validation set of the side of the trot pace in a dressage training at a sampling rate of 50 Hz and a time interval of 2 s, achieving 98.31% accuracy.

**Figure 16 animals-11-02904-f016:**
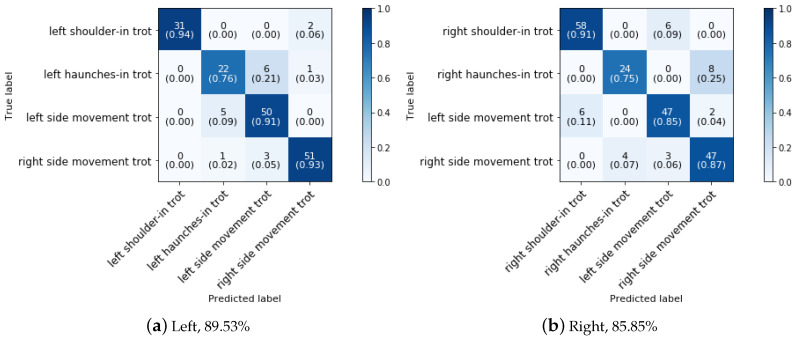
Confusion matrix for the validation set of shouder-in, haunches-in and side movements in trot in a dressage training at a sampling rate of 50 Hz and a time interval of 2 s.

**Figure 17 animals-11-02904-f017:**
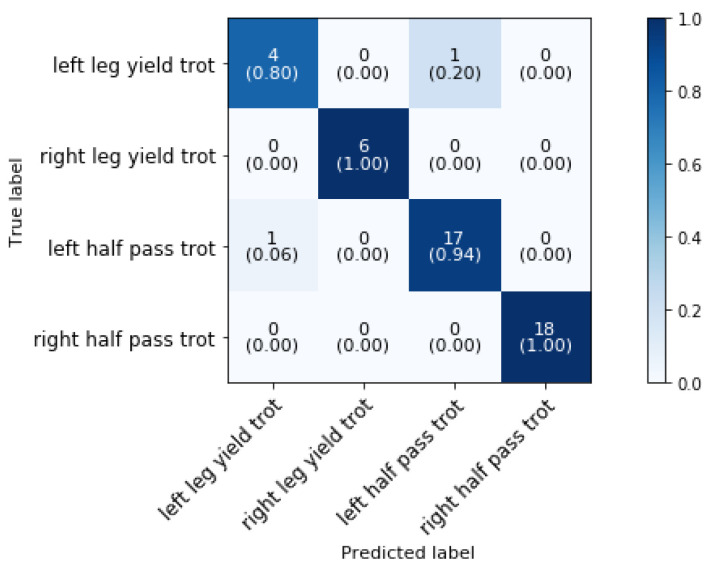
Confusion matrix for the validation set of the trot side movements in a dressage training at a sampling rate of 50 Hz and a time interval of 0.8 s, achieving 95.74% accuracy.

**Figure 18 animals-11-02904-f018:**
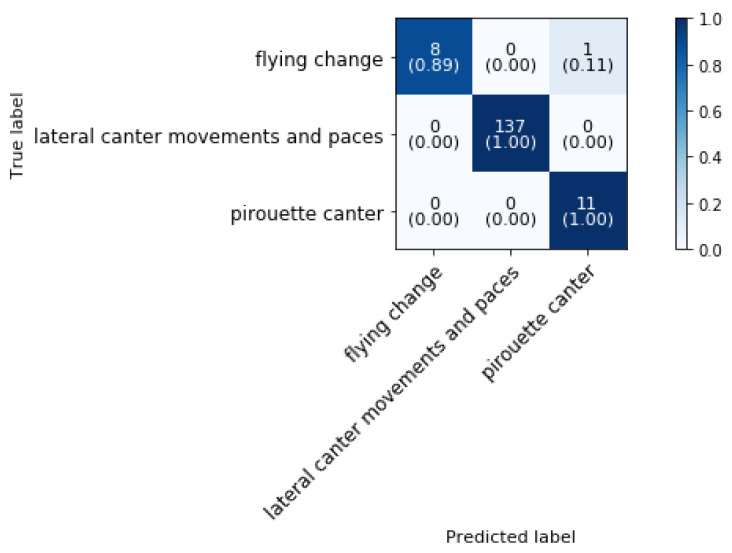
Confusion matrix for the validation set of flying change, lateral canter movements and paces and pirouette canter in a dressage training at a sampling rate of 50 Hz and a time interval of 2 s, achieving 99.36% accuracy.

**Figure 19 animals-11-02904-f019:**
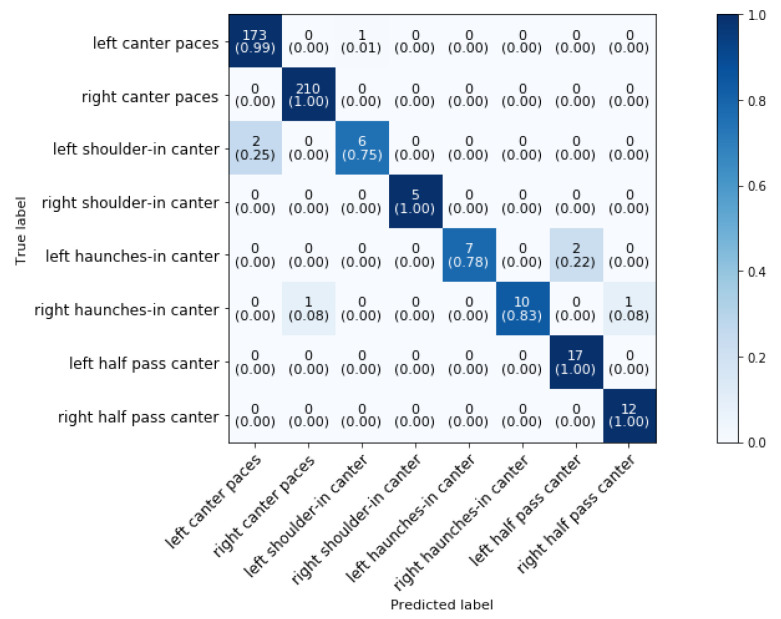
Confusion matrix for the validation set of the canter paces and lateral canter movements in a dressage training at a sampling rate of 50 Hz and a time interval of 0.64 s, achieving 98.43% accuracy.

**Figure 20 animals-11-02904-f020:**
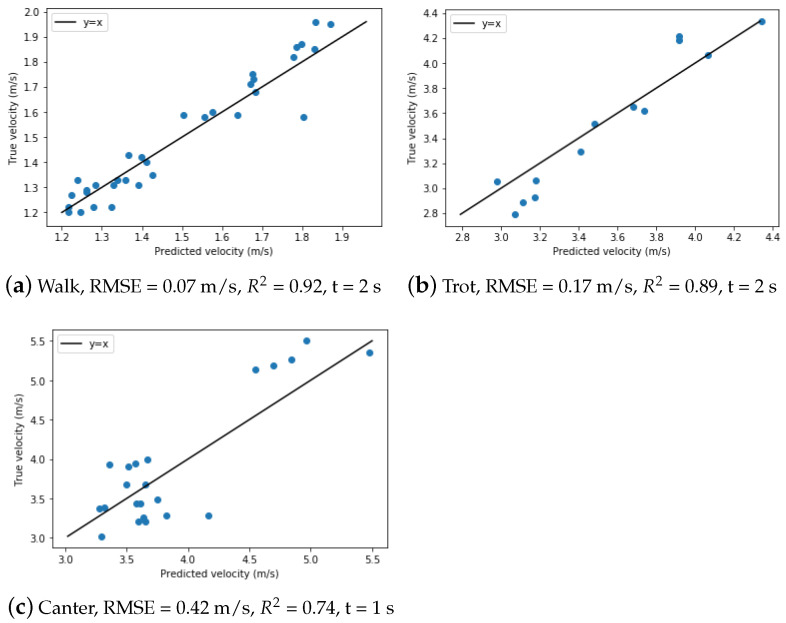
True velocities as function of the estimated velocities of the gaits (walk, trot, canter) with the RMSE, $R^{2}$-value and optimal time interval.

**Figure 21 animals-11-02904-f021:**
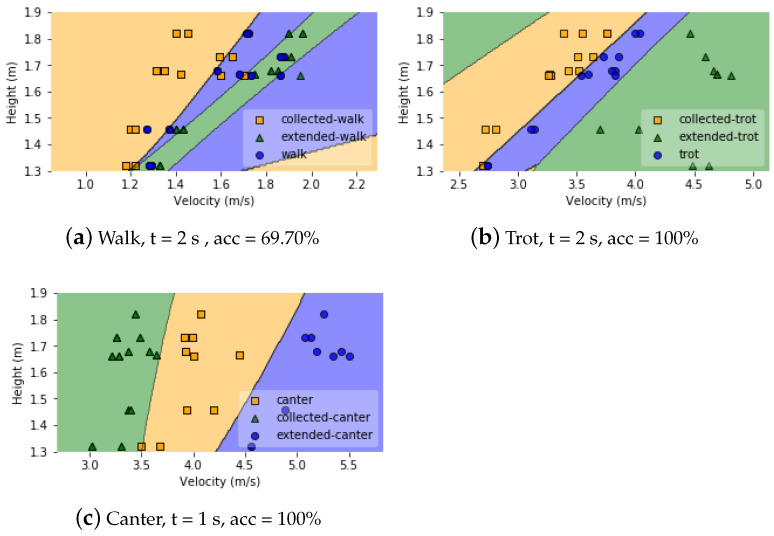
Velocities at collected, normal and extended gaits (walk, trot, canter) as function of the height at withers of the horses together with the decision boundaries of the QDA classifier.

**Figure 22 animals-11-02904-f022:**
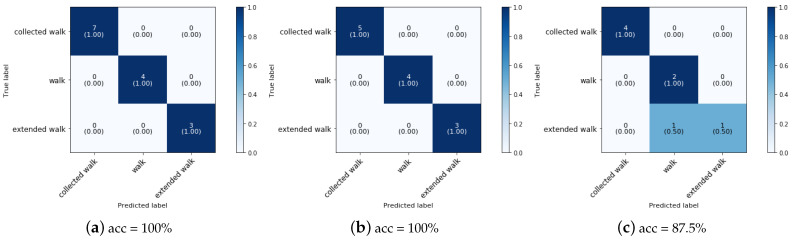
Confusion matrix together with the overall accuracy for the validation set of phase 4 classification of the walk paces with QDA at a sampling rate of 50 Hz and a time interval of 2 s with (**a**) ponies, (**b**) slow horses and (**c**) fast horses.

**Table 1 animals-11-02904-t001:** Title, goal of related work, number of horses in the dataset, sensors and methods proposed in this paper with related work. What makes our research unique is the accurate detection of a the high number of relevant jumping and dressage training activities by automatically extracting features with a CNN using accelerometer data of two legs. NN = Neural Network, DT = Decision Trees, k-NN = k-Nearest Neighbors, NB = Naive Bayes, CNN = Convolutional Neural Network, LDA = linear discriminant analysis, QDA = quadratic discriminant analysis, RF = Random Forest, SVM = Support Vector Machine, LSTM = Long short-term memory, BT = Boosted Trees, GPR = Gaussian Progress Regression.

Paper	Goal	Number of Horses	Sensors	Classification Approach
[5]	Detection of walk, trot and canter	2	Accelerometer	NN, DT, k-NN and NB
[6]	Standing, grazing and ambulating	6	Accelerometer	Threshold based
[11,14]	Detection of stand, walk, trot, canter, roll, paw, flank-watching	6	Accelerometer	CNN
[15]	Detection of stand, walk, trot and canter	20	Accelerometer	Threshold based
[7]	Walk, trot, left canter, right canter, tölt, pace, trocha and paso fino	120	Accelerometer + gyroscope	LDA, QDA, DT, RF, SVM, NN and LSTM
[16]	Estimation of speed in canter	58	Accelerometer + gyroscope	SVM
[17]	Estimation of speed in walk, trot, tölt, pace and canter	40	Accelerometer + gyroscope	SVM, DT, RF, BT, GPR
[18]	Presence/absence and degree of lameness	175	Camera	NN
[4,9,19]	Detection of collected, working, medium and extended pace	6	Camera	Threshold based
[10]	Detection of trot, piaffe and passage	10	Camera	DA
[20]	Gait analysis	35	Strain gauge	NN
[21]	Hoof wall deformation to determine ground reaction forces	1	Strain gauge	NN
[22]	Prediction of load in long bone	9	Strain gauge	NN
[23]	Load-displacement in long bone	13	Strain gauge	NN
This paper	6 jumping and 25 dressage training activities	14	Accelerometer	Hybrid CNN

**Table 2 animals-11-02904-t002:** Participating horses and their height at withers, gender, age, equestrian discipline and level.

Subject Number	Height at Withers (m)	Gender	Age	Equestrian Discipline	Level
1	1.71	Gelding	8	Dressage	Intermediair
2	1.68	Mare	9	Dressage	Intermediair
3	1.68	Mare	7	Dressage	Advanced Medium
4	1.73	Gelding	7	Dressage	Advanced Medium
5	1.73	Mare	9	Dressage	Intermediair
6	1.80	Gelding	14	Dressage	Grand Prix
7	1.82	Gelding	11	Dressage	Grand prix
8	1.66	Mare	7	Dressage	Preliminary
9	1.46	Mare	9	Dressage	Medium
10	1.32	Gelding	9	Dressage	Medium
11	1.66	Gelding	11	Dressage	Grand prix
12	1.76	Gelding	8	Jumping	1.30 m
13	1.71	Gelding	11	Jumping	1.50 m
14	1.75	Gelding	7	Jumping	1.10 m

**Table 3 animals-11-02904-t003:** Observed activities with the proportion and absolute number of samples and the subjects performing the activity. The asterisk (*) denotes movements that are performed in a different gait than the Fédération Equestre Internationale’s (FEI) definition [25].

Equestrian Discipline	Activities				Horses	Proportion (%)	Samples
Dressage				Halt	1, 2, 3, 4, 5, 11	7.56%	16,175
	Walk	Walk paces	Walk paces	Collected walk	2, 3, 4, 5, 7, 8, 9, 10, 11	3.70%	7912
				Working walk	1, 2, 3, 4, 5, 7, 8, 9, 10, 11	22.70%	48,565
				Extended walk	2, 3, 4, 7, 8, 9, 10, 11	4.69%	10,035
		Walk movements		Shoulder-in walk *	5	0.63%	1343
			Lateral walk movements	Haunches-in walk *	3, 5	0.29%	623
				Half pass walk *	3, 5	0.75%	1606
			Pirouette walk	Pirouette walk	2, 3, 4, 5, 11	1.24%	2652
				Collected trot	2, 3, 4, 5, 7, 8, 9, 10, 11	2.78%	5947
		Trot paces	Trot paces	Working trot	1, 2, 3, 4, 5, 7, 8, 9, 10, 11	15.56%	33,281
				Extended trot	2, 3, 4, 5, 7, 8, 9, 10, 11	1.43%	3059
		Trot movements and variations	Variations on the trot gait	Passage	2, 11	2.30%	4927
	Trot			Piaffe	11	0.56%	1197
			Lateral trot movements	Shoulder-in trot	2, 3, 4, 5, 11	2.96%	6325
				Haunches-in trot	2, 3, 4, 5, 11	2.46%	5269
				Leg yield trot	3, 4	1.32%	2814
				Half-pass-trot	2, 3, 4, 5, 11	2.72%	5829
	Canter			Collected canter	1, 2, 3, 4, 7, 8, 9, 10, 11	3.19%	6818
		Canter paces	Canter paces	Working canter	1, 2, 3, 4, 5, 7, 8, 9, 10, 11	14.22%	30,421
				Extended canter	1, 2, 3, 4, 5, 7, 8, 9, 10, 11	1.88%	4012
			Flying change	Flying change	1, 2, 3, 5, 11	2.06%	4404
				Shoulder-in canter *	2, 3	0.63%	1337
		Canter movements and variations	Lateral canter movements	Haunches-in canter	2, 3, 4, 5	0.68%	1447
				Half pass canter	1, 2, 3, 4, 5, 11	1.25%	2666
			Pirouette canter	Pirouette canter	1, 2, 3, 5	2.46%	5254
Jumping	Gaits			Walk	12, 13, 14	29.56%	46,835
				Trot	12, 13, 14	24.98%	39,581
				Left canter	12, 13, 14	20.07%	31,800
				Right canter	12, 13, 14	18.36%	29,091
	Jumping movements			Flying change	13,14	0.32%	503
				Jump	12, 13, 14	6.70%	10,611

**Table 4 animals-11-02904-t004:** Modes of specialization for the classification model, as well as associated recognition accuracies and overall classification accuracy for dressage trainings.

Mode 1	Accuracy (%)	Mode 2	Accuracy (%)	Mode 3	Accuracy (%)	Mode 4	Accuracy (%)	Mode 5	Accuracy (%)
halt	100%	halt	100%	halt	100%	halt	100%	halt	100%
								collected-walk	99.32%
				walk paces	99.32%	walk paces	99.32%	working-walk	99.32%
								extended-walk	86.91%
walk superclass	100%	walk superclass with side	97.67%	walk movements	100%	walk lateral movements	100%	shoulder-in-walk	69.35%
								haunches-in-walk	84.00%
								half-pass-walk	100%
						pirouette-walk	100%	pirouette-walk	100%
								collected-trot	99.48%
				trot paces	99.48%	trot paces	99.48%	working-trot	99.48%
								extended-trot	99.48%
				trot movements	97.04%	variations on the trot gait	97.92%	passage	97.92%
trot superclass	100%	trot superclass with side	93.36%					piaffe	89.02%
						trot lateral movements	96.69%	shoulder-in-trot	87.54%
								haunches-in-trot	71.60%
								leg-yield-trot	78.25%
								half-pass-trot	83.68%
canter superclass	100%	canter superclass with side	100%	canter paces	99.74%	canter paces	99.74%	collected-canter	99.74%
								working-canter	99.74%
								extended-canter	99.74%
				canter movements	97.61%	flying-change	86.77%	flying-change	86.77%
						canter lateral movements	97.61%	shoulder-in-canter	82.60%
								haunches-in-canter	79.02%
								half-pass-canter	97.61%
						pirouette-canter	97.61%	pirouette-canter	97.61%
**Overall accuracy (%)**	**100%**		**97.08%**		**99.10%**		**98.87%**		**96.29%**

**Table 5 animals-11-02904-t005:** Accuracy for the validation set of lateral trot movements in a dressage training at a sampling rate of 50 Hz, a time interval of 2 s and sliding window of 0.5 s using front and back mounted accelerometers (LF = left front, LH = left hind, RF = right front and RH = right hind).

Leg Accelerometers	Accuracy (%)
LF RF	65.15%
LF LH	72.73%
RF RH	80.30%
LH RH	77.27%
LF RH	75.76%
RF LH	84.85%
LF RF LH RH	81.82%

**Table 6 animals-11-02904-t006:** Mean velocities, stride duration and length for each type of gait for horses and ponies (≤148 cm).

Horse Type	Gait	Collected			Normal			Extended		
		(*v*) (m/s)	Δ*T* (s)	(*L*) (m)	(*v*) (m/s)	Δ*T* (s)	(*L*) (m)	(*v*) (m/s)	Δ*T* (s)	(*L*) (m)
Pony	Walk	1.20±0.02	1.32±0.34	1.60±0.43	1.30±0.04	1.12±0.10	1.46±0.14	1.37±0.04	1.08±0.08	1.48±0.16
	Trot	2.73±0.05	0.71±0.04	1.95±0.14	2.93±0.19	0.69±0.03	2.03±0.22	4.21±0.37	0.69±0.05	2.89±0.10
	Canter	3.27±0.15	0.57±0.03	1.87±0.17	3.83±0.26	0.57±0.03	2.20±0.27	4.72±0.16	0.57±0.02	2.67±0.21
Horse	Walk	1.50±0.13	1.30±0.21	1.94±0.34	1.75±0.10	1.26±0.25	2.20±0.41	1.88±0.07	1.19±0.05	2.24±0.15
	Trot	3.46±0.16	0.82±0.02	2.85±0.16	3.80±0.15	0.80±0.01	3.05±0.14	4.64±0.12	0.78±0.03	3.60±0.18
	Canter	3.41±0.14	0.64±0.02	2.18±0.11	4.03±0.16	0.64±0.01	2.57±0.09	5.28±0.14	0.62±0.01	3.26±0.09

## Data Availability

These data are subject to restrictions on their availability. The data was gathered as part of an ongoing project. As such, we do not own this data. We were allowed to publish our work on the sole condition that the data would not be made open source or available to third parties.

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
