# Peer review of "Horse Jumping and Dressage Training Activity Detection Using Accelerometer Data"

_animals, 2021, doi:10.3390/ani11102904_

Round 1

Reviewer 1 Report

The aim of the study was to characterize the horse jumping and dressage training activity detection using accelerometer data.

The approach of the study appears original. The contents of the manuscript are quite interesting by his methodology and through the tools of quantification used. However, I found some parts which should be corrected before an article acceptation.

First of all only 14 horses are used in this study (11 dressage on different level (2-3 in each group; and 3 jumping horses at different level each). In my opinion it is far to low number of animals in each group thus the obtained data is not convincing.

In addition, It will be more suitable if accelerometers will be connected also to the hind tendon in every horse not only in 4 of them.

Introduction

The information about different methods for training monitoring should be added. The most advanced techniques are nowadays used in race horses such as electromyography, peripheral blood mononuclear cell proliferation and activity, infrared thermography or cytokine response.

I suggest to delete the L65-68 because low informative character.

Materials&methods

More precise description of used horses should be added. I am a little bit confused because in L238 it is written that not 3 but 7 showjumping horses have been examined. What do Authors mean by “unseen horses”?

Also some information about examined ponies should be added.

Discussion

There is lack of discussion making the article hard to understand for the reader. Also the part connected with limitations should be added.

Reviewer 2 Report

This paper implements machine learning with the use of horse leg accelerometers to classify horse activity. The paper is very well written and identifies the research methodologies nicely. It can be considered for publication following the suggested below points. These points are required to be clearly addressed in a response letter to the editor/reviewer prior the work to be considered for publication.

The level of literature review in the introduction should be expanded by inclusion of more references on the importance of machine learning and neural networks in equine biomechanics. This must be emphasized in the paper that why researchers should or must use ML and ANN in their study, and not conventional statistical models. What is a hurdle which avoids implementing analytical simulation traditionally adopted to achieve a similar goal? For example, to derive velocity? Examples and reasoning behind such facts are extremely critical, which is however missing in the current paper. The authors can use the reasoning suggested in the following papers and they need to be clearly cited and discussed in the authors’ manuscript, both introduction and discussion:

[1] Schobesberger, H., Peham, C., 2002. Computerized detection of supporting forelimb lameness in the horse using an artificial neural network. The Veterinary Journal 163, 77-84.

[2] Calvert, D., Bajcar, E., Stacey, D., Thomason, J., 2003. Analysis of equine gait through strain measurement, Proceedings of the 25th Annual International Conference of the IEEE Engineering in Medicine and Biology Society (IEEE Cat. No. 03CH37439). IEEE, pp. 2370-2373.

[3] Darbandi, H., Serra Bragança, F., van der Zwaag, B.J., Voskamp, J., Gmel, A.I., Haraldsdóttir, E.H., Havinga, P., 2021. Using Different Combinations of Body-Mounted IMU Sensors to Estimate Speed of Horses—A Machine Learning Approach. Sensors 21, 798.

[4] Savelberg, H., Van Loon, T., Schamhardt, H., 1997. Ground reaction forces in horses, assessed from hoof wall deformation using artificial neural networks. Equine Veterinary Journal 29, 6-8.

[5] Mouloodi, S., Rahmanpanah, H., Burvill, C., Davies, H.M., 2020b. Prediction of load in a long bone using an artificial neural network prediction algorithm. journal of the mechanical behavior of biomedical materials 102, 103527.

[6] Schmutz, A., Chèze, L., Jacques, J., Martin, P., 2020. A method to estimate horse speed per stride from one IMU with a machine learning method. Sensors 20, 518.

[7] Rahmanpanah, H., Mouloodi, S., Burvill, C., Gohari, S., Davies, H.M., 2020. Prediction of load-displacement curve in a complex structure using artificial neural networks: A study on a long bone. International Journal of Engineering Science 154, 103319.

[8] Eerdekens, A., Deruyck, M., Fontaine, J., Martens, L., De Poorter, E., Joseph, W., 2020. Automatic equine activity detection by convolutional neural networks using accelerometer data. Computers and Electronics in Agriculture 168, 105139.

Introduction, lines 59-64 needs to be slightly revised. Make the whole sentence into smaller simpler ones.

Labels for true and predicted classes in figures 8 and 9 should be revised. They are too much outside the box.

A brief explanation in advance of Figure 5 in the results section would make the confusion matrix more easily interpretable for the reader. For example, a concise explanation of difference between diagonal cell (shaded in dark blue) with the off-diagonal cells, true positive rate, true negative rate for the categorizing approach would make the results even more comprehensible for readers new to the ML field (this is commonly the case for researcher in equine biomechanics).

Was there a noise reduction and/or noise addition to virtually be added to the raw datasets prior to training? This should be clearly stated in the materials and Methods.

Was any data normalization implemented prior to network training? Signal processing methods on the raw data?

Sampling rate even up to 2000 Hz is possible with the current advances in IMUs. Was there any specific reason the authors were sampled on a 50 Hz basis?

Line 220: The desired output of the network is velocity. How velocity was extracted from the data? Integration of acceleration or using a GPS to record the speed while gathering the data? If the integration is the case, what about accumulation of errors over time when this approach is conducted? This is referred to as true velocity in Figure 20.

A weak correlation is observed for Figure 20 c. Can the authors explain in the paper and discuss what might be remedy for future research? Also, all subplots in this Figure must be accompanied by R-value, not merely RMSE. Please also draw a shaded line to represent the line X=Y to add visually to the accuracy/level of error in your figure.

A thorough and detailed discussion are missing into how these results and the significance of this conducted study is adding to literature. The authors are expected to use literature (some examples of references were mentioned earlier in this review report) and make connections with those studies and see how this paper is adding to knowledge already published. Please refer back and forth to section/figures/tables of your paper to those claims/suggestions/discussions already discussed in the literature. This needs to be emphasized both in introduction and discussion of the paper.

While quite significant results are presented, drawn conclusions are trivial. The authors are recommended to expand more and stive to emphasize the essence of their study.

Round 2

Reviewer 1 Report

Dear Authors,

I am glad that You have corrected the article. Now the quality of presented work is much higher. Still the number of horses is low, however, as Authors mentioned similar works have been published recently.

Thus, I reccomend the article for publication in Animals.

Author Response

We would like to express our gratitude to the reviewer for the thoughtful comments and efforts to improve our paper. Additionally, we thank the reviewer for the positive evaluation.

Reviewer 2 Report

I am pleased to advise that the authors have implemented necessary changes in a satisfactory manner and the paper can be accepted for publication.

However, some minor comments. The English in the section "related work" where the authors have added the new references needs to be polished, at least during the production stage. For example, " [5] and [18] employ neural networks with features as input." This is an example of a very poor scientific statement, both in its structure and in its meaning. There are more such statements that the reviewer is not going to mention all. Lastly, not all the suggested and similar publications were added to the manuscript.

Author Response

We would like to express our gratitude to the reviewer for his or her thoughtful suggestions and efforts to enhance our paper. As requested by the reviewer, we have polished the related work section. Additionally, we've incorporated all of the references mentioned by the reviewer.